

# Managing Data of Sensor-Equipped Transportation Networks using Graph Databases

Erik Bollen[1,2], Rik Hendrix[2], and Bart Kuijpers[1]

[1]Databases and Theoretical Computer Science Group, Data Science Institute (DSI), Hasselt University and transnational University Limburg, Agoralaan building D Diepenbeek 3590, Belgium
[2]Data Science Hub, VITO, Boeretang 200 Mol 2400, Belgium

**Correspondence:** Erik Bollen (erik.bollen@uhasselt.be)

**Abstract.** In this paper, we are concerned with data pertinent to *transportation networks*, which model situations in which objects move along a graph-like structure. We assume that these networks are equipped with *sensors* that monitor the network and the objects moving along it. These sensors produce *time-series data* resulting in sensor networks. Examples are river-, road- and electricity networks.

Geographical information systems are used to gather, store and analyse data, and we focus on these tasks in the context of data emerging from transportation networks equipped with sensors. While tailored solutions exist for many contexts, they are limited for sensor-equipped networks at this moment. We view time-series data as temporal properties of the network and approach the problem from the viewpoint of property graphs. In this paper, we adapt and extend the theory of the existing property graph databases to model spatial networks, where nodes and edges can contain temporal properties that are time-series data originating from the sensors. We propose a language for querying these property graphs with time series, in which time-series and measurement patterns may be combined with graph patterns to describe, retrieve and analyse real-life situations. We demonstrate the model and language in practice by implementing both in Neo4j and explore questions hydrology researchers pose in the context of the Internet of Water, including salinity analysis in the Yser river basin.

## 1 Introduction

Transportation networks are a common research subject. These networks range from river networks that transport water and road networks transporting vehicles to heat or electricity networks that transport energy. A Transportation network is characterised by having a stable topology, that is, the connectivity between nodes does not change often, and objects or substances move through the topology. The objects or substances are measured or tracked by some sort of sensors, and these sensors create time-series data. Using graphs to model transportation networks is a common practice based on the connection information often rising from vector-based GIS data. Because these sensors themselves do not move, the time series can be attributed to the nodes and edges representing the vector elements. In the networks, there are also static data, which can be represented as common properties in the graph, such as the IDs of the vector elements. As a result, researchers have to process property graphs with time series in the nodes and edges. This is different from the field of temporal graphs because nodes and edges are always considered present here. Only the properties can change through time, in which case, a property is considered to have





different values at different points in time. In contrast, most work about temporal graphs assumes that nodes and edges can be removed or added, resulting in graphs where node and edge are linked with validity time intervals.

With the rise of the Internet of Things, the number of sensors in transportation networks has grown considerably, and the rate at which these sensors produce data is increasing too McCabe et al. (2017). Therefore, the time series can have a high resolution of data points. Nodes and edges are well-supported in graph databases, and query languages exist to process them.

However, managing time series in graph databases is not trivial. Additional data structures are required to store the time series, and the existing query languages do not support time series to express constraints. Similarly, time-series databases are well established with their query languages to deal with the timestamps and values. However, adding the graph structure of the network is not supported. A system that fully integrates and supports both graph structures and time series data is missing.

In this work, we use an existing model for property graphs to create a property graph model that includes time series in nodes

and edges. We leverage pattern matching, commonly used in the current graph language, to create a query language and logic that can answer queries over the proposed graph model. One crucial part is that, in the query language the temporal properties, with their timestamp-value pairs, can interact with the nodes, edges, and their static properties. The natural order present in the time series is built into the query language to simplify writing constraints in the queries. This provides tools for studying and transforming the data for advanced analysis. The proposed logic is well suited to objectively describe and study problems or

shortcomings in the model or query language. Next to the theory in this work, the usage of the model and and implementation is demonstrated on a practical use-case within the Internet of Water project where electrical conductivity data is analysed on the Flemish Yser river.

A very preliminary version of the theory, presented in this paper, was published in Bollen Bollen (2022), without theoretical framework and implementation. We want to use the present work to develop the theoretical foundation and to show how it can

be implemented. In addition, we discuss experiments and additional examples here.

In Section 2, we discuss the related work and in Section 3 the theoretical model for property graphs with time series is given. Section 4 defines the logic for the proposed query language and shows how this language can be realised using the Graph Pattern Matching Language (GPML). The section ends with some application examples. In Section 5, we describe our implementation of the graph model and query language using Neo4j and Cypher together with an experiment based on the

Internet of Water project. Section 6 and Section 7 discuss the proposed model and query language and present our conclusion.

## 2   Existing Work

There is a long-lasting overlap between geographical information systems and the field of data management, especially if physical networks are studied. Our sensor networks closely align with the definition of "Geosensor Networks" by Nittel (2017). She points out that the data is often handled by domain scientists who need to work with the data, preferably without needing





to know the lower-level details of data management. This is also demonstrated in practice by the research of Rodríguez-Alarcón and Lozano (2022) for river basins, Hornsby and King (2008) for traffic on road networks, and Gilbert et al. (2018) for electricity networks, where each network is a transportation network in our definition and is modelled using a graph. Especially in the case of studying rivers, a lot of work is happening in Belgium, where climate change has a big impact because of the increasing occurrences of extreme meteorological events, Brouwers et al. (2015). It is, therefore, not surprising to see parties

involved in handling rivers to join forces and invest in full-stack approaches encompassing measuring, storing, analysing and managing the data, as in the Internet of Water project[1]. In this context, colleagues also studied the river Scheldt, Bollen et al. (2023), but approached the transportation network by modelling it as an interval labelled graph. In this work, we want to focus more on the raw time-series data in the geographical context.

In recent years, two types of graphs, "Resource Description Framework" (RDF) graphs and "property graphs", have taken

the foreground. For the latter, the recent definition used is the definition given by Angles (2018). Bonifati et al. (2018) give a slightly different definition of property graphs. Before that, other versions of graphs were used, for example, the weighted graph Devienne and Lebegue (1986). George and Shekhar (2018) use this model and define Time Aggregated Graphs. In a Time Aggregated Graph, the weight of nodes and edges can vary over time. An example is the travel time in a network. This is part of dynamic, or temporal, graphs where the study focuses on nodes and edges that exist during a certain time Debrouvier

et al. (2021). There are also advanced systems, for example, GRADOOP by Rost et al. (2022), that incorporate scaling and many interaction possibilities. This field is still an active research domain, as the recently started project "HyGrpah"[2] shows. Even though existing temporal graphs might be able to accommodate our case, a dedicated graph definition for a property graph where certain properties are time series is not available to the best of our knowledge.

One of the first publications regarding the underlying logic or calculus for graph query languages is a query language

on graphs with recursion by Cruz et al. (1987) from 1987. In this work, the relational model is used to store the data. An approach based on a graph model is called regular path expressions and is described by Backofen (1993). These ideas have been extended to regular path queries in tandem with the evolution of the graph model itself, Angles and Gutiérrez (2008). Since then, different papers have been published that further build on regular path queries. Angles et al. (2017) show how this all fits together in the query languages. In their vision, the two basic elements of the graph query language are graph patterns

and, secondly, navigational patterns. The evolution is summarised by Bonifati et al. (2018) as the existence of Regular Path Queries (RPQ) and Conjunctive Graph Queries (CQ), which, when taken together, is the language of Conjunctive Regular Path Queries (CRPQ). Adding disjunction to CRPQ results in a more expressive query language called Union Conjunctive Regular Path Queries (UCRPQ). When considering edge labels in two directions, the theory also talks about Two-Way Regular Path Queries (2RPQ), Conjunctive Two-Way Regular Path queries C2RPQ and Union C2RPQ, Barceló Baeza (2013). However,

these language classes do not account for reasoning over the properties of graphs. To accommodating this, the class of Regular

---

[1]https://www.internetofwater.be/wat-is-internet-of-water/
[2]https://hygraph.net/



Property Graph Queries can be used, based on Regular property graph Logic and Regular property graph Algebra, Bonifati et al. (2018).

Despite the history, practical query languages have only been implemented in recent years. In contrast to early graph inter-action, which was mainly imperative API-like, recent query languages are declarative. The most important existing languages
are Cypher (OpenCypher and GQL), SPARQL, Gremlin, and GSQL. There is an effort to standardise the graph query languages, as it was done for SQL. This new standard, called GQL, incorporates elements from (open)Cypher, G-core, PGQL and Tigergraph's GSQL[3]. As Deutsch et al. (2022) show, these languages, especially GQL and SPARQL, have a common base for matching graph patterns, and they call it Graph Pattern Matching Language, GPML. The theoretical foundation of GPML is recently described by Francis et al. (2022). They provide a calculus that uses the same concepts as the Regular Property Graph
Logic.

For time-series data, declarative time-series query languages and imperative query approaches exist. An elaborate theory, as in the case of the graph databases, does not exist to the best of our knowledge. The first versions of the time-series database InfluxDB[4] have the query language InfluxQL, which uses SQL-like syntax and describes the results of the query in a declarative format. Now, a new language is used, called Flux, which is an imperative language where time series form a source in
transformations. In an extended version of PostgreSQL, using TimescaleDB[5], time series can be queried using SQL. However, they treat time series as tabular data, and they do not exploit the order of the measurements in the time series induced by the natural order of time.

## 3    A Data Model for Property Graphs with Time Series

The property graph model is well suited to model transportation networks, and various definitions of this model exist. For
example, the property graph model definition of Angles (2018) and the definition of Bonifati et al. Bonifati et al. (2018). In this work, the latter is used, because it aligns better with our applications. Our definition of the *Property Graph with Time Series* model adapts the definition of Bonifati et al. in the book "Querying Graphs" Bonifati et al. (2018) and adds time series based on the idea of Llusà Serra et al. (2016).

**Definition 3.1.** Given are a set of values $\mathcal{V}$, a finite set of labels $\mathcal{L}$ and a set of timestamps $\mathcal{T}$ (equipped with a total (temporal)
order $\leq$). Further, we assume that a set of property keys $\mathcal{K}$ is given, with two disjoint subsets $\mathcal{K}_S$ ($S$ stands for static) and $\mathcal{K}_T$ ($T$ stands for temporal). The sets $\mathcal{V}, \mathcal{L}, \mathcal{T}$ and $\mathcal{K}$ are assumed to be pairwise disjoint. Lastly, the set $\mathcal{M} \subseteq \mathcal{T} \times \mathcal{V}$ denotes the set of measurements which are $(timestamp, value)$-pairs.

A **property graph with time series**, is then defined to be a structure

$$G = (\mathcal{N}, \mathcal{E}, \lambda, \upsilon_S, \upsilon_T),$$

---

[3]https://www.gqlstandards.org/existing-languages
[4]https://www.influxdata.com/products/influxdb/
[5]https://www.timescale.com





where:

- – $\mathcal{N}$ is a finite set of nodes;

- – $\mathcal{E}$ is a set of directed edges, where each edge belongs to $\mathcal{N} \times \mathcal{N}$;[6]

- – $\lambda : \mathcal{N} \cup \mathcal{E} \to \mathcal{P}_{<\omega}(\mathcal{L})$ is a partial function assigning to nodes and edges a finite set of labels (here, $\mathcal{P}_{<\omega}(\mathcal{L})$ denotes the set of all finite subsets of $\mathcal{L}$);

- – $v_S : (\mathcal{N} \cup \mathcal{E}) \times \mathcal{K}_S \to \mathcal{V}$ is a partial function assigning a value to a static property of nodes and edges; and

- – $v_T : (\mathcal{N} \cup \mathcal{E}) \times \mathcal{K}_T \to \mathcal{P}_{<\omega}(\mathcal{M})$ is a partial function assigning a finite set of measurements to a temporal property of nodes and edges. We require that in such a set, no timestamp appears twice, and we call the image of the function $v_T$ a *time series*. □

Now, we introduce our simplified fictitious running example of such a graph, which is used throughout this paper.

**Example 3.1.** Our example of a property graph with time series contains seven nodes that all have a numeric identifier from $1$ to $7$, and these make up the set of nodes $\mathcal{N}$. In addition, there are edges that represent the connectivity of the network. The connectivity is chosen to be unidirectional to reflect a river-like transportation network. There are six edges in total, being $(1,3), (2,3), (3,4), (4,5), (6,5)$, and $(6,7)$. A visual representation is shown in Figure 1. Each edge is identified by a pair of node identifiers. All nodes and edges can have zero or multiple labels to categorise them. In this example, we label all nodes with the label `point`, and we label all edges with the label `path`. Each node and edge is assigned a name to demonstrate the static properties. For the nodes, this name corresponds to the letter "N" together with the identifier, and for the edges "E" with a number. For example, node 1 has a static property with key `name` and value `N1`, thus $(1, \text{name})$ is mapped to `N1`. The edge $(1,3)$, together with `name`, is mapped to the value `E1`. All mappings for the static properties are listed below. The temporal properties form the last part of the example. They consist of a property key and a value. The property value is considered to be a time series, which is a set of $(timestamp, value)$-pairs. The values in the time series can be almost anything, but in this example, we use numerical values. For example, for node 1 the mapping is

$$(1, \texttt{water-level}) \mapsto$$
$$\{(\texttt{2022-08-15T10:00}, 14), (\texttt{2022-08-15T11:00}, 14),$$
$$(\texttt{2022-08-15T12:00}, 13), (\texttt{2022-08-15T13:00}, 13),$$
$$(\texttt{2022-08-15T14:00}, 12), (\texttt{2022-08-15T15:00}, 12)\},$$

which means that there is a time series with six values on August 15th, 2022 around noon. Next to the nodes, there are also edges that have a temporal property, there the times series represents the travel-time for an object travelling along that edge.

---

[6]In the original work of Bonifati et al. Bonifati et al. (2018), these two sets are disjoint subsets of a bigger object set. We simplify this approach by not considering this object set. In addition, our directed edges are defined by an ordered pair of nodes, because of which the function $\eta$ is not needed.



We remark that, in this example, the timestamps are behaving perfectly. That is, timestamps are spread evenly, and all time series use the same timestamps. This does not need to be, probably nor will it be in reality. However, the example is easier to understand by assuming these characteristics.

The complete formal description of the example is as follows, with the remark that in the time series, the timestamps are on 2022-08-15, but only the time is shown for readability.

– $\mathcal{N} = \{1,2,3,4,5,6,7\}$;

– $\mathcal{E} = \{(1,3),(2,3),(3,4),(4,5),(6,5),(6,7)\}$;

– $\lambda = \{1 \mapsto \{\texttt{point}\}, 2 \mapsto \{\texttt{point}\}, 3 \mapsto \{\texttt{point}\}, 4 \mapsto \{\texttt{point}\}, 5 \mapsto \{\texttt{point}\}, 6 \mapsto \{\texttt{point}\}, 7 \mapsto \{\texttt{point}\},$
    $(1,3) \mapsto \{\texttt{path}\}, (2,3) \mapsto \{\texttt{path}\}, (3,4) \mapsto \{\texttt{path}\}, (4,5) \mapsto \{\texttt{path}\}, (6,5) \mapsto \{\texttt{path}\}, (6,7) \mapsto \{\texttt{path}\}\}$;

– $\upsilon_S = \{(1,\texttt{name}) \mapsto \texttt{N1}, (2,\texttt{name}) \mapsto \texttt{N2}, (3,\texttt{name}) \mapsto \texttt{N3}, (4,\texttt{name}) \mapsto \texttt{N4}, (5,\texttt{name}) \mapsto \texttt{N5}, (6,\texttt{name}) \mapsto \texttt{N6},$
$(7,\texttt{name}) \mapsto \texttt{N7}, ((1,3),\texttt{name}) \mapsto \texttt{E1}, ((2,3),\texttt{name}) \mapsto \texttt{E2}, ((3,4),\texttt{name}) \mapsto \texttt{E3}, ((4,5),\texttt{name}) \mapsto \texttt{E4},$
    $((6,5),\texttt{name}) \mapsto \texttt{E5}, ((5,7),\texttt{name}) \mapsto \texttt{E6}\}$;

– $\upsilon_T = \{$
    $(1,\texttt{water-level}) \mapsto \{(\texttt{10:00},14), (\texttt{11:00},14), (\texttt{12:00},13), (\texttt{13:00},13), (\texttt{14:00},12), (\texttt{15:00},12)\},$
    $(2,\texttt{water-level}) \mapsto \{(\texttt{10:01},15), (\texttt{11:02},15), (\texttt{12:01},15), (\texttt{13:01},15), (\texttt{14:00},15), (\texttt{15:01},16)\},$
$(3,\texttt{water-level}) \mapsto \{(\texttt{10:00},14), (\texttt{10:30},13), (\texttt{11:00},14), (\texttt{11:30},14), (\texttt{12:00},14), (\texttt{12:30},15),$
    $(\texttt{13:00},14), (\texttt{13:30},14), (\texttt{14:00},13), (\texttt{14:30},12), (\texttt{15:00},13)\},$
    $(5,\texttt{water-level}) \mapsto \{(\texttt{10:00},14), (\texttt{11:00},14), (\texttt{12:00},15), (\texttt{13:00},15), (\texttt{14:00},14), (\texttt{15:00},14)\},$
    $(6,\texttt{water-level}) \mapsto \{(\texttt{10:00},18), (\texttt{11:00},18), (\texttt{12:00},17), (\texttt{13:00},18), (\texttt{14:00},19), (\texttt{15:00},19)\},$
    $(7,\texttt{water-level}) \mapsto \{(\texttt{10:00},18), (\texttt{11:00},17), (\texttt{12:00},16), (\texttt{13:00},17), (\texttt{14:00},18), (\texttt{15:00},18)\},$
$((3,4),\texttt{travel-time}) \mapsto \{(\texttt{10:00},3.10), (\texttt{11:00},3.50), (\texttt{12:00},3.49), (\texttt{13:00},3.47), (\texttt{14:00},3.46),$
    $(\texttt{15:00},3.44)\},$
    $((4,5),\texttt{travel-time}) \mapsto \{(\texttt{10:00},3.20), (\texttt{11:00},3.15), (\texttt{12:00},3.53), (\texttt{13:00},3.51), (\texttt{14:00},3.48),$
    $(\texttt{15:00},3.46)\}\}.$

This concludes our example, of which a visual representation is shown in Figure 1.                                   □

The total temporal order $\leq$ on the set $\mathcal{T}$, assumed by Definition 3.1, induces a total order on the measurements in a time series. Indeed, the temporal order $\leq$ on $\mathcal{T}$ induces an order (which we also denote by $\leq$) on $\mathcal{M}$, as follows: for $m_1, m_2 \in \mathcal{M}$, we define $m_1 \leq m_2$ if and only if $\tau(m_1) \leq \tau(m_2)$, where $\tau : \mathcal{M} \to \mathcal{T}$ is the projection on the time component (that is, $\tau$ returns the timestamp of a measurement). Furthermore, we define $\nu(m)$ to be the value of the measurement $m$ (that is, its second component). In other words, a measurement $m \in \mathcal{M}$ is the couple $(\tau(m), \nu(m))$. The above order allows us to define a
"next"-relationship and a "previous"-relationship between measurements in a time series. The definition of both is as follows.





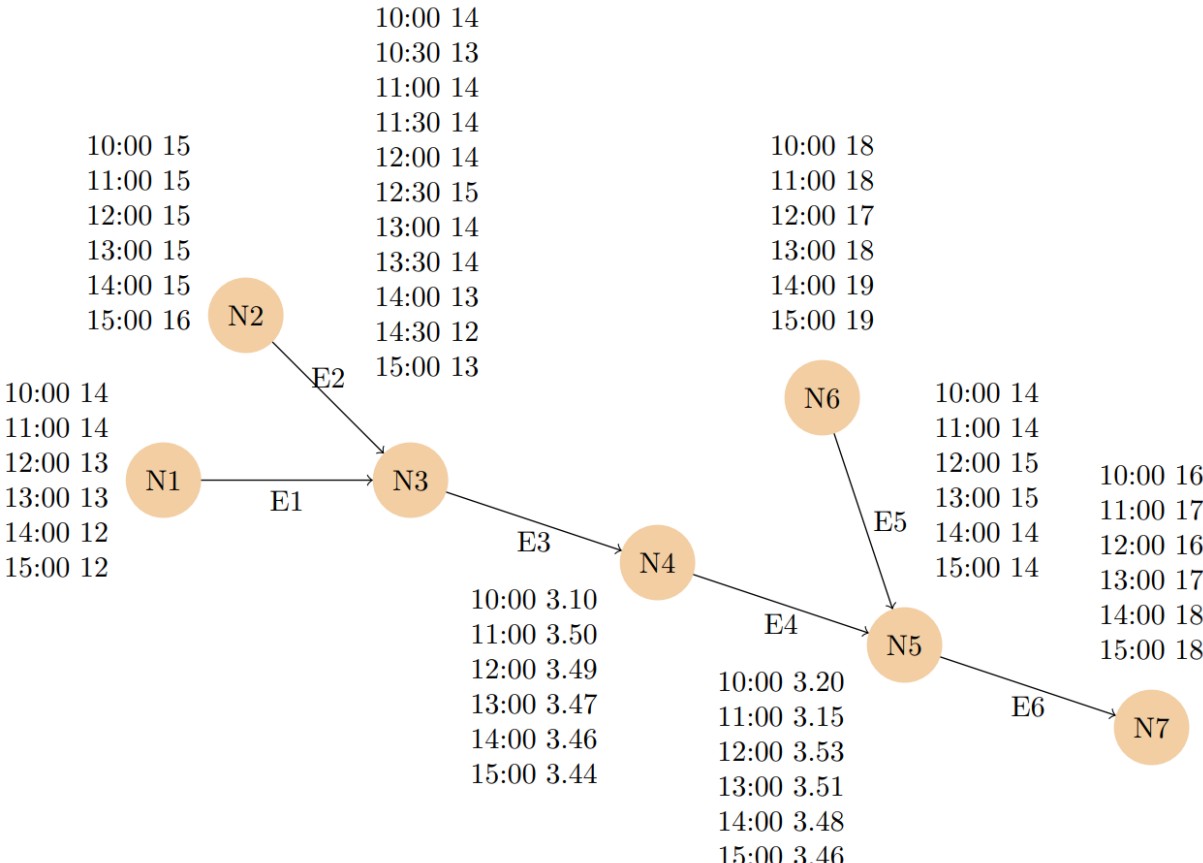

**Figure 1.** Visual graph representation of Example 3.1. The nodes are assigned a label `point` and the edges are labelled `path`, these labels are not shown. The timestamps also include a date, 2022-08-15, but it is not shown to reduce the clutter.

Given two measurements $m_1$ and $m_2$ in a time series $s$, $m_2$ is the next measurement after $m_1$, denoted $next(m_2, m_1, s)$ if and only if $m_1, m_2 \in s$, $m_1 < m_2$, and there is no measurement $m$ in $s$ with $m_1 < m < m_2$ (here, $<$ has the obvious meaning of $\leq$ but not equal). The definition for the previous measurement, $previous(m_1, m_2, s)$, then corresponds to $next(m_2, m_1, s)$. In the case that the second condition is dropped (that is, there might exist another measurement between the other two), then we use predicates $after(m_2, m_1, s)$ to indicate that $m_2$ occurred after $m_1$ and $before(m_2, m_1, s)$ for the other case. In addition, we can extend this idea to define the first and last measurements. The $first(m, s)$ and $last(m, s)$ predicates are true for a measurement $m$, within a time series $s$, where there is no previous or next measurement for $m$ in $s$, respectively.





## 4   A Formal Query Language for Property Graphs with Time Series

### 4.1   Extending Regular Property Graph Logic

The concept of Regular Path Queries is well established and is an important concept Nolé and Sartiani (2016)Abiteboul and Vianu (1999)Libkin and Vrgoč (2012). For example, Cypher and GQL are based on the theory of Regular Property Graph Queries, described by Bonifati et al. Bonifati et al. (2018). These queries can be described using a Regular Property Graph Logic or a Regular Property Graph Algebra. To obtain a query language for property graphs with time series, we extend the Regular Property Graph Logic with elements to incorporate the time series. More specifically, we add the description of time-

series patterns for a temporal property in a node or an edge. The result is that the temporal properties are handled just as the nodes, edges and properties. A query can contain node, edge, path, static property, and temporal properties constraints that can be evaluated simultaneously. These constraints may depend on each other without the need for multiple queries.

We take the syntax and semantic definitions of Regular Property Graph Logic by Bonifati et al. Bonifati et al. (2018), and with it, define our logic for Property Graph Query Language with Time Series, or GQL-TS Logic. This is done by adding formal

descriptions for temporal properties and querying them. We show how these logical elements can be realised as time-series and measurement patterns in the Graph Pattern Matching Language. In the last part, example queries are given to demonstrate the different possibilities of our Graph Query Language with Time Series, which we abbreviate as GQL-TS.

### 4.1.1   Syntax of GQL-TS Logic

In this section, we give the syntactical definition of GQL-TS Logic.

A query, expressed in GQL-TS Logic, is conceived to take a property graph with time series (as presented in Definition 3.1) as input and it takes the form of a set of non-recursive Datalog-style rules. Each rule is a description of a sub-graph pattern or path of the input graph, and is of the form

$$head \leftarrow body_1, ..., body_m, constraint_1, ..., constraint_n \tag{1}$$

which has a head predicate and, in the body, a sequence of body predicates, possibly in combination with additional constraints.

First, we need some additional notions and notations. We use $\mathcal{N}^{var}$ to denote the (possibly infinite) set of variables that represent nodes in a graph. Similarly, we have the set $\mathcal{E}^{var}$ of variables that refer to edges in a graph. Finally, we assume to have an infinite set $\mathcal{L}^{(2)}$ of fresh predicate names of arity two.

We define the components of a query rule in GQL-TS Logic. First, we define the form of the "head" of a rule, which either creates a new relation (called $\ell_2$) or a result relation of some arity $a$.

**Definition 4.1.**   The $head$ of rule (1) has either the form

$$\ell_2(x, y)$$



where $\ell_2 \in \mathcal{L}^{(2)}$, $x,y \in \mathcal{N}^{var}$, or the form

$$\texttt{result}(x_1,...,x_a)$$

with $a \geq 1$ and $x_1,...,x_a \in \mathcal{N}^{var}$. In the first case, we define $var(head) := \{x,y\}$ and in the second case, we define $var(head) := \{x_1,...,x_a\}$. □

Next, we define the form of the "body" components of a rule, which either uses an existing or previously created edge relation (called $\ell_2$); computes the transitive closure of such an edge relation (denoted by $\ell_2^*$); or uses an existing unary node relation (called $\ell_1$).

**Definition 4.2.** Each $body_i$ in rule (1) has the form

– $\ell_2(x,y)$ `AS` $e$;

– $\ell_2^*(x,y)$; or

– $\ell_1(x)$,

where $\ell_2 \in \lambda(\mathcal{E}) \cup \mathcal{L}^{(2)}$ is of arity two and is not the predicate name used in the head of the rule, $\ell_1 \in \lambda(\mathcal{N})$ is of arity one, $x,y \in \mathcal{N}^{var}$ and $e \in \mathcal{E}^{var}$. In the first case, we define $var(body_i) := \{x,y,e\}$, in the second case, we define $var(body_i) := \{x,y\}$ and in the third case, we have $var(body_i) := \{x\}$. □

Next, we define the form of the "constraint" components of a rule and we distinguish between "static constraints" and "time-series constraints". In this definition, we need a set $\mathcal{M}^{var}$ of measurement variables that range over measurements. When $m \in \mathcal{M}^{var}$, then $m.value$ is a variable that ranges over values in $\mathcal{V}$ and $m.time$ is a variable that ranges over timestamps in $\mathcal{T}$. Finally, we have a set $\Theta_v$ of binary operators (or relations) on the set of values $\mathcal{V}$, and a set $\Theta_t$ of binary operators on $\mathcal{T}$, to which the total order on timestamps $\leq$ is assumed to belong. Typically, we take $\Theta_v$ and $\Theta_t$ to be $\{=, \neq, \leq, <, \geq, >\}$.

**Definition 4.3.** Each $constraint_j$ in rule (1) is either a *static constraints* or a *time-series constraints*.

(1) *Static constraints* are of the form

– $x.\rho_1 \; \theta_v \; y.\rho_2$;

– $x.\rho_1 \; \theta_v \; val$; or

– $x = y$,

where $x,y \in \mathcal{N}^{var} \cup \mathcal{E}^{var}$, $\rho_1, \rho_2 \in \mathcal{K}_S$, $val \in \mathcal{V}$, and $\theta_v$ a binary operator from $\Theta_v$. We define $var(constraint_j) = \{x,y\}$, for the first and third case, and $var(constraint_j) = \{x\}$ in the second case.

(2) *Time-series constraints* are of the form

– $x.\sigma = [m_1,...,m_p]$, with $p \geq 1$;





- $x.\sigma = [m_1, ..., m_k, *, m_{k+1}, m_p]$, with $p \geq 2$ and $1 \leq k < p$;

- $v_1 \; \theta_v \; v_2$; or

- $t_1 \; \theta_t \; t_2$,

where $x \in \mathcal{N}^{var} \cup \mathcal{E}^{var}$, $\sigma \in \mathcal{K}_T$ and $m_1, ..., m_p \in \mathcal{M}^{var}$ and where, furthermore,

- $\theta_v \in \Theta_v$, $v_1, v_2$ are

    - values from $\mathcal{V}$;

    - of the form $m.value$, with $m \in \mathcal{M}^{var}$; or

    - the result of a function (with co-domain in $\mathcal{V}$) applied to value- and time constants and variables; and

- $\theta_t \in \Theta_t$, $t_1, t_2$ are

    - timestamps from $\mathcal{T}$,

    - of the form $m.time$, with $m \in \mathcal{M}^{var}$, or

    - the result of a function (with co-domain in $\mathcal{T}$) applied to value- and time constants and variables. □

We are ready to give the definition of a rule in GQL-TS Logic.

**Definition 4.4.** A **GQL-TS Logic rule** has the form

$$head \leftarrow body_1, ..., body_m, constraint_1, ..., constraint_n$$

with $m > 0$ and $n \geq 0$, where $head$, $body_i$ and $constraint_j$ are as defined in Definitions 4.1, 4.2 and 4.3, with the additional restriction that

$$var(head) \subseteq \bigcup_{i=1}^{m} var(body_i) \cup \bigcup_{j=1}^{n} var(constraint_j).$$

□

A query expressed in GQL-TS Logic is then simply described by a (non-recursive) collection of GQL-TS Logic rules. At least one rule in a query needs to have a special head `result` describing the final output of the query. We need to discuss the notion of the "dependency graph" of query $Q$ such that we can define what is meant by non-recursive. In the rules of $Q$, several predicates may appear: some are existing node and edge predicates of the property graph with time series; some are newly created predicates. In the "dependency graph of $Q$", the predicates are nodes, and there is an edge from predicate $\ell$ to predicate $\ell'$ if $\ell$ appears in the head and $\ell'$ in the body of some rule in $Q$. When $\ell$ appears in the head and $\ell'$ in the body of some rule, we call $\ell'$ a "successor" of $\ell$ and $\ell$ a "predecessor" of $\ell'$. This dependency graph is called non-recursive if it is acyclic and recursive if it contains a cycle. With that, we can give the formal definition of a GQL-TS Logic query.





**Definition 4.5.** A **GQL-TS Logic query** is a finite non-empty and non-recursive set of rules such that at least one rule has a head predicate `result` and all `result` predicates have the same arity. □

When the predicate `result` appears more than once, we assume that this results in the unions of the rules, as will be clear in the next section, where we define the semantics of GQL-TS Logic.

### 4.1.2   Semantics of GQL-TS Logic

In this section, we give the semantics of GQL-TS Logic queries when applied to a property graph with time series. Two parts form the semantics: (a) the bodies and constraints of rules are satisfied by an assignment, and (b) each valid assignment

contributes elements to a set defined by the head of a rule. The first part will be called the satisfaction of rules, and the second part will be the evaluation of rules. We first define the rule evaluation and then discuss the satisfaction problem for the bodies and constraints.

Given a property graph with time series $G = (\mathcal{N}, \mathcal{E}, \lambda, \upsilon_S, \upsilon_T)$, a query $Q$, expressed in GQL-TS Logic, consists of a finite set rules $R_Q$, where a rule $r \in R_Q$ is a GQL-TS Logic rule and where at least one rule in $R_Q$ has $\texttt{result}(x_1, ..., x_a)$ as its

head. We denote the predicate of the head of rule $r$ by $head(r)$.

When a rule $r$ has head predicate $\ell = head(r)$, and only when the semantics of all predicates $\ell'$ that are successors of $\ell$ in the dependency graph of $Q$ have been evaluated on $G$, then we can define the semantics of the predicate $\ell$ on $G$. We remark that this recursive process stops since the dependency graph of $Q$ is acyclic.

The evaluation of a rule $r \in R_Q$ on $G$ results in the creation of an $c_r$-ary relation, where $c_r$ is the arity of $\ell = head(r)$. This $c_r$-ary relation is defined in terms of assignments of the form

$$\mu : \mathcal{N}^{var} \cup \mathcal{E}^{var} \cup \mathcal{M}^{var} \to \mathcal{N} \cup \mathcal{E} \cup \mathcal{M},$$

that assign identifiers to node, edge and measurement variables. Suppose that rule $r$ has the form $head \leftarrow body_1, ..., body_n,$ $constraint_1, ..., constraint_m$, then we define $Sat(r)$ to be the set of assignments $\mu$ for which

$$G \models (body_1 \wedge \cdots \wedge body_m \wedge constraint_1 \wedge \cdots \wedge constraint_n)[\mu]$$

in the usual sense used in predicate logic (for our specific constraints, in particular those on time series, we give the details below). If the head of $r$ has the form $\ell(x_1, ..., x_{c_r})$, then the evaluation of the rule $r$ on $G$ is the set

$$[[r]]_G := \{(\mu(x_1), ..., \mu(x_{c_r})) \mid \mu \in Sat(r)\}.$$

The head $\ell(x_1, ..., x_{c_r})$ of rule $r$ may appear in several rules of the query $Q$ (possibly with other variable names). Then, we define the semantics of this predicate $\ell$ in $G$ to be

$$[[\ell]]_G := \bigcup_{r \in R_Q, head(r) = \ell} [[r]]_G.$$





The result of the evaluation of the query $Q$ on the property graph with time series $G$ is then defined to be the $a$-ary relation

$$[[Q]]_G := \bigcup_{r \in R_Q, head(r)=\texttt{result}} [[r]]_G.$$

What remains to be specified is the meaning of $G \models body_i[\mu]$ and $G \models constraint_j[\mu]$, for a $body_i$ in a rule $r$ (as in Definition 4.2) and $constraint_j$ in a rule $r$ (as in Definition 4.3).

**Satisfaction of Body Predicates** We define the meaning of $G \models body_i[\mu]$ for each of the various forms of $body_i$, that we defined earlier in Definition 4.2.

A body predicate $body_i$ in a rule $r$ is satisfied by a mapping $\mu$:

– if $body_i$ is of the form $\ell_2(x,y)$ AS $e$ and

     – $\ell_2 \in \lambda(\mathcal{E})$, then $G \models (\ell_2(x,y)$ AS $e)[\mu]$ if there exists an edge in $\mathcal{E}$, such that $\mu(e) = (\mu(x), \mu(y)) \in \mathcal{E}$ and $\ell_2 \in \lambda(\mu(e))$; or

     – $\ell_2 \in \mathcal{L}^{(2)}$, then $G \models (\ell_2(x,y)$ AS $e)[\mu]$ if there exists an edge in $[[\ell_2]]_G$ with $\mu(e) = (\mu(x), \mu(y)) \in [[\ell_2]]_G$.

– if $body_i$ is of the form $\ell_2^*(x,y)$, an additional set is needed, before we can define the satisfaction condition. For this purpose, we introduce the following notation

$$x \xrightarrow{\ell, G} y,$$

which expresses that the tuple $(x,y)$ belongs to (1) the transitive closure of the edges in $G$ with label $\ell$, when $\ell$ is an existing label in $G$; (2) the transitive closure of a newly created relation, labelled $\ell$, when $\ell$ is not in the original graph $G$. The set $[[\ell_2^*]]_G$ is then defined as

$$\{(x,x) \mid x \in \mathcal{N}\} \cup \{(x,y) \mid x,y \in \mathcal{N} \text{ and } x \xrightarrow{\ell_2, G} y\}.$$

We want to remark that the meaning of this set has to be interpreted as "zero or more times an edge with label $\ell$". Therefore, the pairs $(x,x) \mid x \in \mathcal{N}$ are included to ensure that assignments where the edge occurs zero times are possible. Finally, $G \models (\ell_2^*(x,y))$ if $(\mu(x), \mu(y)) \in [[\ell_2^*]]_G$.

– if $body_i$ is of the form $\ell_1(x)$ with $\ell_1 \in \lambda(\mathcal{N})$, then $G \models (\ell_1(x))$ if $\mu(x) \in \mathcal{N}$ and $\ell_1 \in \lambda(\mu(x))$.

**Satisfaction of Constraints** We define the meaning of $G \models constraint_{i_j}[\mu]$, for each form of $constraint_j$, that appears in Definition 4.3. For each of the comparison relations $\theta_v$ and $\theta_t$, we also use this symbol for their interpretation in the structure $G$ (abusing notation).

– If $constraint_j$ is of the form $x.\rho_1 \ \theta_v \ y.\rho_2$, then $G \models (x.\rho_1 \ \theta_v \ y.\rho_2)[\mu]$ if $\mu(x), \mu(y) \in \mathcal{N} \cup \mathcal{E}$ and $\upsilon_S(\mu(x), \rho_1) \ \theta_v \ \upsilon_S(\mu(y), \rho_2))$;





– If $constraint_j$ is of the form $x.\rho_1 \ \theta_v \ val$, then $G \models (x.\rho_1 \ \theta_v \ val)[\mu]$ if $\mu(x) \in \mathcal{N} \cup \mathcal{E}$ and $\upsilon_S(\mu(x), \rho_1) \ \theta_v \ val$;

– If $constraint_j$ is of the form $x = y$, then $G \models (x = y)[\mu]$ if $\mu(x) = \mu(y)$.

For the constraints concerning the time series, two additional functions are assumed to exits. We assume a function exists, with co-domain in $\mathcal{V}$, that takes as input value- and time constants and variables. We denote this function with $f_\mathcal{V}$. Similarly, the function taking value- and time constants and variables with co-domain $\mathcal{T}$ is represented with $f_\mathcal{T}$.

– If the form of $constraint_j$ is $x.\sigma = [m_1, ..., m_p]$, then $G \models (x.\sigma = [m_1, ..., m_p])[\mu]$ if:

  – $\mu(x) \in \mathcal{N} \cup \mathcal{E}$,

  – $\upsilon_T(\mu(x), \sigma) = \{\mu(m_1), ..., \mu(m_p)\}$,

  – and for all $m_l$ with $1 \le l \le p$:

    • $\mu(m_l) \in \mathcal{M}$, and

    • $next(\mu(m_{l+1}), \mu(m_l), \upsilon_T(\mu(x), \sigma))$ holds.

We remark that if $p = 1$, then $next()$ can not be satisfied and should be ignored (similarly, when $l = p$).

– If the form of $constraint_j$ is $x.\sigma = [m_1, ..., m_k, *, m_{k+1}, m_p]$, with $p \ge 2$ and $1 \le k < p$, then $G \models (x.\sigma = [m_1, ..., m_k, *, m_{k+1}, m_p])$ if:

  – $\mu(x) \in \mathcal{N} \cup \mathcal{E}$,

  – $\upsilon_T(\mu(x), \sigma) = \{\mu(m_1), ..., \mu(m_p)\}$,

  – and for all $m_l$ with $1 \le l \le p$:

    • $\mu(m_l) \in \mathcal{M}$, and

    • $next(\mu(m_{l+1}), \mu(m_l), \upsilon_T(\mu(x), \sigma))$ holds, except for $l = k$, then $after(\mu(m_{l+1}), \mu(m_l), \upsilon_T(\mu(x), \sigma))$ holds.

– If $constraint_j$ has the form $v_1 \ \theta_v \ v_2$, and:

  – $v_1$ is of the form $m.value$ and $v_2 \in \mathcal{V}$, then $G \models (v_1 \ \theta_v \ v_2)[\mu]$ if $\mu(m) \in \mathcal{M}$ and $\nu(\mu(m)) \ \theta_v \ v_2$;

  – $v_1$ is the result of $f_\mathcal{V}(m.time)$ with $m \in \mathcal{M}^{var}$ and $v_2 \in \mathcal{V}$, then $G \models (v_1 \ \theta_v \ v_2)[\mu]$ if $\mu(m) \in \mathcal{M}$ and $f_\mathcal{V}(\tau(\mu(m))) \ \theta_v \ v_2$;

  – $v_1$ and $v_2$ are value constants, the result of $f_\mathcal{V}$ applied to time constants, or the result of $f_\mathcal{V}$ after $\mu$ applied on variables, then $G \models (v_1 \ \theta_v \ v_2)[\mu]$ if $v_1 \ \theta_v \ v_2$.

– If $constraint_j$ has the form is $t_1 \ \theta_t \ t_2$, and

  – $t_1$ is of the form $m.time$ and $t_2 \in \mathcal{T}$, then $G \models (t_1 \ \theta_t \ t_2)[\mu]$ if $\mu(m) \in \mathcal{M}$ and $\tau(\mu(m)) \ \theta_t \ t_2$;

  – $t_1$ is the result of $f_\mathcal{T}(m.value)$ with $m \in \mathcal{M}^{var}$ then, $G \models (t_1 \ \theta_t \ t_2)[\mu]$ if $\mu(m) \in \mathcal{M}$ and $f_\mathcal{T}(\nu(\mu(m))) \ \theta_t \ t_2$;

  – $t_1$ and $t_2$ are value constants, the result of $f_\mathcal{T}$ applied to value constants, or the result of $f_\mathcal{T}$ after $\mu$ applied on variables, then $G \models (t_1 \ \theta_t \ t_2)[\mu]$ if $t_1 \ \theta_t \ t_2$.





## 4.2 Graph Query Language with Time Series

In this section, the GPML notation is extended with elements to describe time series with time-series patterns consisting of measurement patterns. Important is that, in a time-series pattern, two consecutive measurement patterns implicate that the measurements matched need to fulfil the $next()$ relationship. First, the syntax of a measurement pattern is described. Afterwards, we demonstrate how these can be combined into a time-series pattern, and finally, these time-series patterns are linked with existing GPML notation.

### 4.2.1 Measurement Pattern

We consider a measurement a (timestamp, value)-pair. A measurement pattern `<m>` describes a measurement, where `m.timestamp` represents the timestamp component and `m.value` the value component. For example, if `m` is matched to `(2022-08-15T13:00,17)`, then `m.timestamp = 2022-08-15T13:00` and `m.value = 17`.

### 4.2.2 Time-Series Pattern

A time series consists of one or multiple measurements with an order on their timestamp. Therefore, a time-series pattern is one or multiple measurement patterns that describe a subset of the time series together. Two concatenated measurement patterns express two measurements for which the $next$ predicate holds. This means, for the pattern `<m><n>` is series $s$, $next(n, m, s)$ is true (or $previous(m, n, s)$).

In GPML, some quantifiers allow patterns to be repeated. The Kleene star is the most generic but is accompanied by three additional quantifiers. These quantifiers can be used in the time-series patterns. However, the meaning is slightly different. Between two measurements, `m` and `n`, an additional, anonymous measurement can be matched if the Kleene star precedes the second measurement pattern, for example, `<m>*<n>`. The repetition can be limited by a minimum `*{a,}`, that is, the pattern has to occur at least `a` times, or by a range `*{a,b}`, in which case the pattern has to occur at least `a` times and not more than `b` times. The last quantifier is `+`, which is equal to `*{1,}`.

### 4.2.3 Time-Series Patterns as Temporal Properties

Time-series patterns match temporal properties that belong to nodes or edges. The same notation as for the static properties is used to link the time-series patterns to a graph pattern. To recap, we will first show some important notations of GPML. We refer to Deutsch et al. (2022) for a complete description of the notation. A pattern in GPML describes the structure to which sub-graphs need to adhere to be considered a match. Additional information, such as properties and labels, are constraints added to the pattern.

A node is described by a set of round brackets, a possible name and a label within the brackets, separated by a colon. Property constraints can be expressed after the `WHERE` clause. The property key is written, followed by a constant or variable that constrains the property value. This is demonstrated in the following example.





```
(a:point WHERE name = 'N1');
```

The pattern describes a node with label `point` and a property `name` that needs to have the value `N1`. Each node that matches the pattern is assigned to the variable `a`. We remark that, `a` represents one node match at a time.

An edge is represented by two square brackets and an arrow. The pattern can contain a label, an edge type, a variable name, and properties. The notation for these three elements is exactly the same as they are for node patterns.

```
()-[b:path WHERE name = 'E3']->();
```

In this example, an edge is described with a type `path`, and it must have a property `name`, which should be equal to `E3`. The direction of the edge is indicated by the arrow to the right. The two `()` patterns describe the existence of a node with no additional requirements, but any valid node pattern is allowed. Similar to the empty nodes, the most basic edge pattern between two nodes is `-[]-`, which only requires the existence of an edge with no additional constraints. Together with two blank nodes, `()-[]-()`, the pattern would produce all matches of two nodes with an edge in between them. An edge can be matched zero or multiple times by using the Kleene star or one of the other repetition notations.

Finally, a query consists of node- and edge patterns, preceded with the keyword `MATCH`, and followed by an optional keyword `WHERE`, after which additional constraints can be written using the variables that are used in the node- and edge patterns.

To express a temporal property of a node or edge, a time-series pattern needs to be added to the node or edge properties, together with the name of the temporal property. This is expressed by writing after the `WHERE` keyword, the temporal property key, then a colon, followed by a time-series pattern. To differentiate the static and temporal properties, both groups are separated by the keyword `SERIES`, and we will always place the static properties first.

#### 4.2.4 Examples

The following are queries for the running example, given in Example 3.1 and shown in Figure 1.

**Example 4.1.** In this example, the query's result is the temperature at two o'clock in node with the name `N6`. The corresponding pattern matches node with `name` equal to `N6` and a `water-level` property, which is a temporal property, with one measurement pattern `x`. The timestamp of measurement `x` needs to be equal to `2022-08-15T14:00`.

```
MATCH (n:point WHERE name = "N6" SERIES water-level = <x>)
WHERE x.timestamp = "2022-08-15T14:00";
```

There is only one match in the example graph, that is, node `6` with at two o'clock the value 19.

**Example 4.2.** In this second example, two different nodes are matched, requiring both to have a temporal property `water-level`, in which a measurement with the same value at the same time exists.





```
MATCH (n:point WHERE SERIES water-level = <a>),
      (m:point WHERE SERIES water-level = <b>)
WHERE a.value = b.value
      AND a.timestamp = b.timestamp
      AND NOT n = m;
```

In the example graph, this query will produce 8 matches. One where n maps to node 1 and m maps to node 3 and where the value is 14 at 10:00 and 11:00. Secondly, one where m is mapped to 5 for 10:00 and 11:00 with a value 14. Another one, with n → 3 and m → 5 with at 10:00 and 11:00 the value 14. The last match is for nodes 2 and 5 with value 15 at 12:00 and 13:00.

We remark that there are 16 matches. The same match can be made for each match given with the nodes reversed for n and m. These 8 additional matches are not shown. This query can prevent the double results by requiring that the ID of node n is smaller than that of node m.

**Example 4.3.** The following example focuses on variable length paths and the variable length time-series patterns.

```
MATCH (n:point WHERE name = "N7" SERIES water-level = <a>)
    <-[:path+]-(m:point WHERE SERIES water-level = <b>*<c>
WHERE a.timestamp = "2022-08-15T15:00"
      AND a.timestamp = c.timestamp
      AND b.value >= a.value;
```

The query should match all nodes upstream of node 7 that contain a temporal property water-level, wherein a measurement exists that occurred before 15:00 and where the value of that measurement is higher than the value in node 7 at 15:00. Valid matches are where m is equal to node 6 and where measurement b is assigned to (14:00,19), (13:00, 18), (11:00, 18), or (10:00, 18). We remark that the measurement of 15:00 in node 6 is not matched because the *previous*() predicate still needs to be true for measurements b and c. Similarly, node 7 is not matched with n and m because + is used in the edge pattern, which means the edge should occur at least once.

The temporal properties are treated at the same level as static properties, nodes, and edges.[7] This means the dependency between temporal and static properties can be expressed within one query. The following example depicts a situation where node or edge properties determine the time a temporal property needs to be constrained.

**Example 4.4.** In this example, the query matches measurements based on values that occurred earlier in the path we want to match. The time that the travel-time series are matched depends on the measurements in the node. Further, down the path, it again depends on the previous matches.[8]

---

[7]We refer to section 5 that shows an equivalence between temporal properties and nodes and edges in the graph.
[8]There is more to be said about the equality of time stamps in time series. One can not easily assume that the exact same timestamp is present in two different time series. However, for now, this is assumed.





```
MATCH (a:point {water-level=<m1>})
    -[x:path {travel-time=<t1>}]->(b:point {water-level=<m2>})
    -[y:path {travel-time=<t2>}]->(c:point {water-level=<m3>})
WHERE t1.value > threshold
    AND t1.time = m1.time
    AND m2.time = m1.time + t1.value
    AND t2.time = m2.time
    AND m3.time = m2.time + t2.value;
```

Extending this example to include static properties in the constraints is possible. This could be, for example, when only the speed is given, and the length of the edge is needed to calculate the travel time. We want to remark that taking the sum of two values is not described in the logic, nor are other operations on values. We assume they can be used as long as the result is again a valid value. For example, in current Cypher implementations, such operations do exist.

Compared to procedures implemented in a query language or methods implemented using drivers for the databases, the method presented here gives more possibilities for writing constraints. In addition, early pruning of the graph or time-series matching is now possible. If they are treated separately, the graph part needs to be retrieved first, and subsequently, the time-series data needs to be queried, or vice versa. This does not provide the same flexibility as our method.

## 5  Implementation

To demonstrate the idea in practice, we developed a database system that can store graphs as defined in Definition 3.1 and query it with our proposed query language. Based on the additions described in section 4.2, an existing query language is extended to facilitate queries considering the new temporal properties. We chose to realise this with the Neo4j database. It is well-established in practice and provides a good foundation. The open-source query parser from OpenCypher, used in Neo4j, provides a suitable starting place for the query-language implementation. Adding other functionalities to Neo4j is possible by using user-defined functions and procedures.

### 5.1  Storing in Practice

To store the time series in the graph database, we build the concept of the *Full Graph Model*. In this approach, the graph database is used to store the network's topology and time series. The latter are represented as linked lists of nodes in the graph. For each measurement, a node containing the timestamp and value of the measurement is created. These measurements can be linked by edges, connecting all measurements in the temporal order. This means that for each time series, there is a linked list of nodes where each node is a measurement. A head time series node is added to these lists when the time series is linked to a node (of the spatial part). This way, the node of the topology and the head nodes can be linked by a specially typed edge. For edges, this is not possible. There, a similar link can be established by adding a unique ID in the head node of the time series and





the static properties of the edge. This would also be possible for nodes, leading to a more consistent design. But we asses that it would impact the query performance, although future research has to verify this. We chose to link the oldest measurement as the first measurement to the head node. This means the latest measurement is stored at the end of the linked list. We selected this approach because, this way, the order of measurements described in the queries corresponds to the order in which they are matched. It could be interesting to store the most recent measurement first because users might be more interested in recent measurements. Of course, the Full Graph model has a noticeable impact on the total number of nodes and edges stored, as we will show. However, practical experiments have not yet shown any limitations.

## 5.2 Querying in Practice

A new query parser was implemented using Antlr[9] to implement the query language. This parser can parse the Graph Query Language with Time Series and translate the queries into standard Cypher. This is possible because time-series and measurement patterns are actual graph patterns. Specifically, it is possible to translate each measurement pattern into a node pattern and a time-series pattern into a path pattern. These translated patterns can be joined with the pattern for the topology in one query, which can be evaluated by the default query engine of Neo4j. This query translator is implemented with a Spring Boot server that takes a query, sends the translation to the Neo4j database, and returns the result. The user can interact with this system using a web interface implemented using React.

Our grammar for the query language is based on the G4 Cypher grammar. The original grammar specification can be found on the OpenCypher git repository[10]. The grammar describes valid measurement and time-series patterns by adding the following rules.

```
oC_SeriesPatterns :
    SERIES SP oC_SeriesPattern SP? (',' SP? oC_SeriesPattern)* ;

oC_SeriesPattern :
    oC_PropertyKeyName SP? ':' SP? oC_MeasurementPattern+ ;

oC_MeasurementPattern :
    oC_RangeLiteral?
    oC_LeftArrowHead SP?
    (oC_Variable | (oC_Expression SP? ',' SP? oC_Expression))
    SP? oC_RightArrowHead ;

SERIES :
```

---

[9]https://www.antlr.org
[10]It was deleted on October 17th 2022, commit 38b5e39). We use the version available on the repository on 27 July 2022.





```
( 'S' | 's' ) ( 'E' | 'e' ) ( 'R' | 'r' )
( 'I' | 'i') ( 'E' | 'e' ) ( 'S' | 's' ) ;
```

To link these patterns with the existing property patterns, only the rule expressing the mapping for properties needs to be updated to include the time-series pattern.

```
oC_MapLiteral:
  '{' SP?
  (oC_PropertyKeyName SP? ':' SP? oC_Expression SP?
(',' SP? oC_PropertyKeyName SP? ':' SP? oC_Expression SP?)* )?
  (oC_SeriesPatterns)?'}' ;
```

Compared to the description with GPML, the properties of nodes and edges are not expressed after a "WHERE" clause but are enclosed in curly brackets. In addition, equals signs linking property keys and property values (for static and temporal properties) are replaced by a colon. This nicely demonstrates that the description of our query language with GPML is easily
implemented in any GPML-based query language. We want to remark that recently, a new specification version was made available, and our changes are not yet compatible with this new version. Our full grammar specification is available on OSF[11].

### 5.3 Experiment

This section focuses on implementing the proposed theory and testing its feasibility. An experiment was conducted to study water-quality measurements in the Internet of Water project as proof of concept.

In Flanders, the "Internet of Water" project[12](IoW) aims to enhance monitoring and governance of the Flemish waterways. Hundreds of sensors are deployed and monitored along the Flemish rivers. The sensors' data is made available on the data platform of the Flemish environmental agency, which is one of the partners in the project. VITO (Flemish Institute for Technological Research) contributes to the project by analysing the data Pagán et al. (2020) to answer questions such as "What is the salinity at any possible part of the river?" and "What is the current drought status of a river?". These questions are important
to scientists and managers since climate change leads to more extreme meteorological weather phenomena. These situations affect water supply and water quality, for example, due to the influence of the salty sea on rivers, which can harm the drinking water and surrounding land area, Gobin (2012).

In this set-up, we study the Yser river west of Flanders, depicted in Figure 2. It is a prone region to salt intrusion because it is close to the sea. In dry periods, salty seawater will push land inwards along the river and intrude via ground water, Desmet
and Bauwel (2023). For this reason, 42 electrical-conductivity (EC) and water-temperature (WT) sensors have been placed on the river, and we use the resulting measurements to pose typical questions.

---

[11]DOI: 10.17605/OSF.IO/J9CN5, https://osf.io/j9cn5/
[12]https://www.internetofwater.be/wat-is-internet-of-water/





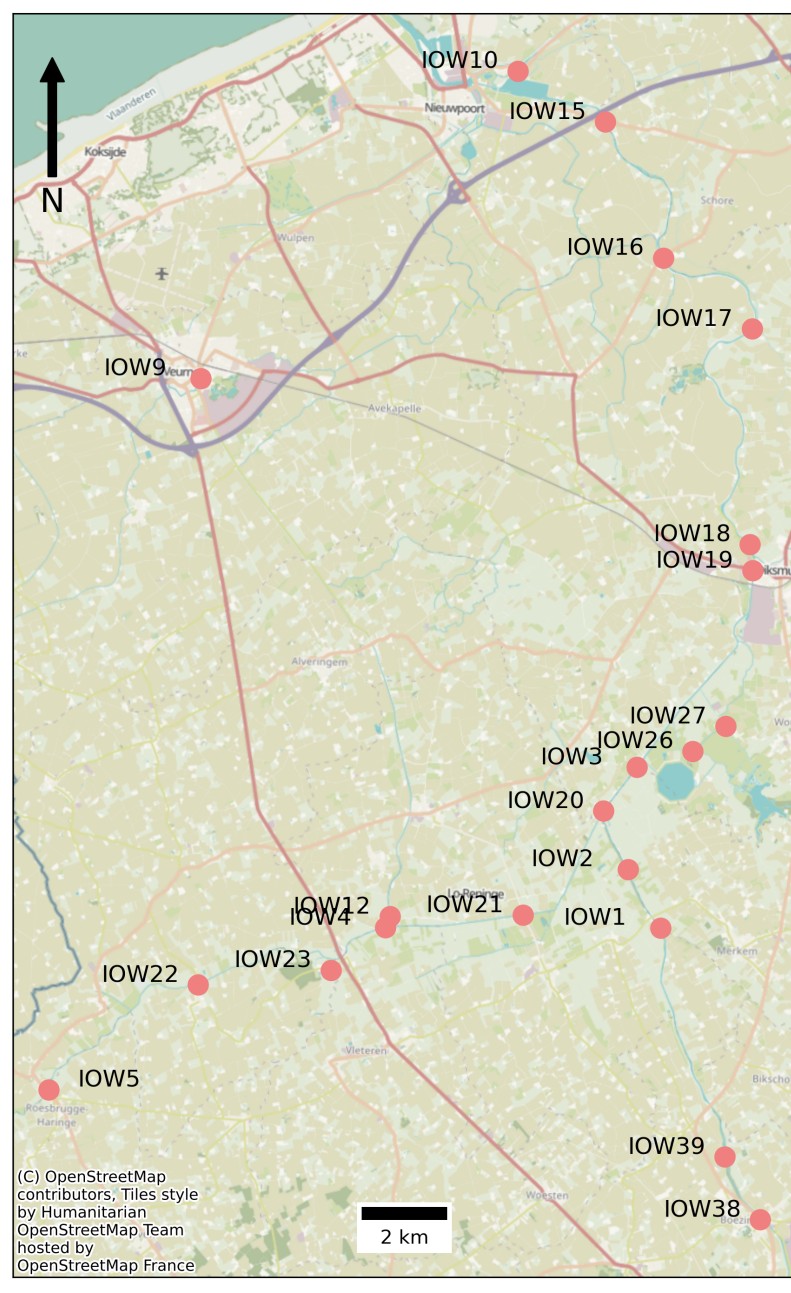

**Figure 2.** Overview of the Yser river in the west of Belgium. Only a subset of the stations are shown. There are more stations in the data set, but they are not drawn to reduce clutter.





**Table 1.** Disk usage of the Neo4j database for the Yser data.

| Node storage | Edge storage | Properties storage | Total storage |
|---|---|---|---|
| 5.93 MB | 13.49 MB | 32.53 MB | 53.04 MB |

To model the river, we use the Vlaamse Hydrographic Atlas (VHA), a digital GIS data set representing all rivers in Flanders. In this data set, the rivers consist of smaller river segments, represented as line geometries. Sensors are attributed to segments, and because the researchers are interested in studying the river per segment, we chose to model each segment as a node in the graph. If water flows from one segment to another, the two nodes, representing those two segments, are connected by an edge indicating the water flow. In total, there are 534 segments in this use case. The graph contains thus 534 nodes, and they are connected by 541 edges. Distributed over this network are 42 sensors, each producing EC measurements every 15 minutes[13]. From now on, when values of the measurements are shown, they are in micro Siemens per centimetre, $\mu S/cm$. We selected the data between the first of January 2022 and the first of May 2022. One time series has 4 months of data with a resolution of 15 minutes, which results in 11520 measurements per sensor. However, since it is real life and some measurements are missing, not all series do have so many measurements. With the measurements added, the database using the *Full Graph Model* consists of 395 074 nodes. This results in Neo4j (version 4.2.3) in 53 MB disk space usage. Details are included in Table 1.

Before we discuss the queries in detail, some additional information about the use case is needed. Most of the sensors in the Internet of Water are numbered and named, for example, "IOW1" for Internet of Water sensor number 1. Some sensors have slightly different names because they already existed before the project. This name can be used to identify the series of the sensor in the graph uniquely. Our implementation allows users to add labels to the series, as it is possible for nodes and edges. For this use-case, all series are EC25 measurements, so we assign the label "ec25" to each series. To identify a series by name or label, the following notation is used: `name:label:pattern`. For example, `IOW1:ec25:<a>` is the pattern for a series with the name "IOW1" and the label "ec25". N additional constraints on the measurement a. Similarly, we can use the pattern `IOW1::<a>` for the series with the name "IOW1" and no label requirements, or `:ec25:<a>` for a series that has the label "ec25" with no name requirements. This addition is not yet described or defined in our proposed Logic but is implemented in the proof of concept. We will study the data by posing different questions researchers within the Internet of Water pose. These queries will be increasingly complex.

**Query 1** Every segment is uniquely identified with the *vhas* number (the id of the segment). In this first query, we are going to look up the location of a sensor by querying for the *vhas* of the segment. The `DISTINCT` keyword ensures that only one row is returned. Because every measurement in the series can be matched with the pattern, the query without `DISTINCT` would return as many rows as measurements in the series. The result of this query is one record containing the value `6033646` for `n.vhas`.

```
MATCH (n:segment {SERIES IOW1:ec25:<a>}) RETURN DISTINCT n.vhas;
```

---

[13]Waterinfo.be already takes into account the water temperature when providing the EC values.





**Table 2.** Result table containing first ten records from the answer to Query 3.

| n.vhas | a.value |
|--------|---------|
| 6042697 | 7441.90 |
| 6018926 | 2899.00 |
| 6033571 | 777.16 |
| 6021226 | 3705.00 |
| 6021185 | 1470.00 |
| 6033487 | 751.89 |
| 7073733 | 2388.43 |
| 6033467 | 10362.00 |
| 6033467 | 7842.00 |
| 6033615 | 8597.00 |
| ... | ... |

**Query 2** The second example query is a typical question where the value of a specific moment and location is requested. In this case, the value of the IOW 1 sensor at midnight on March 30th.

```
MATCH (n:segment {SERIES IOW1::<a>})
WHERE a.timestamp = datetime("2022-03-30T00:00:00+0000")
RETURN a.value;
```

The value, that is `a.value`, is 1230 for the series of IOW1 at the specified timestamp. The result of the query contains only one record because there is only one match in the graph for this pattern.

**Query 3** With example three, we show how a label can access a series with a specific type. The names or locations might be unknown, but the series type is known. Here, we want to access EC measurements and see their values on March 30th at midnight. The results are shown in Table 2 where, next to the value, the *vhas* is returned of the segment where the series is on.
In total 36 records are returned but we truncated the table for readability reasons.

```
MATCH (n:segment {SERIES :ec25:<a>})
WHERE a.timestamp = datetime("2022-03-30T00:00:00+0000")
RETURN n.vhas, a.value;
```

**Query 4** This query demonstrates a more advanced series pattern. Because consecutive measurement patterns match con-
525 secutive measurements in the series, it is possible to describe a peak. A peak is described as three measurements `a`, `b` and `c` where the value of `b` is higher than the value of `a` and `c`.





**Table 3.** Five first records retrieved by Query 4 showing 5 peak values in sensor IOW1.

| b.timestamp | b.value |
| --- | --- |
| 2022-01-01T01:00:00Z | 536.72 |
| 2022-01-01T02:00:00Z | 536.17 |
| 2022-01-01T06:30:00Z | 557.84 |
| 2022-01-01T07:30:00Z | 557.15 |
| 2022-01-01T11:15:00Z | 550.97 |

```
MATCH (n:segment {SERIES IOW1:ec25:<a><b><c>})
WHERE a.value < b.value AND c.value < b.value
RETURN DISTINCT b.timestamp, b.value LIMIT 5;
```

The result of this query, where only the first five matches are returned, is shown in Table 3. With each peak, we also return the timestamp of measurement b to see when the peak occurred. Although this query matches many small local peaks, it demonstrates the declarative power of time-series patterns. More advanced peak descriptions can be built to find peaks of interest more precisely.

**Query 5** This query demonstrates how a surpass of a threshold in a series can be identified. Such a query interests scientists
who, for example, want to find river pollution spills. They know that the EC value would pass a threshold in the event of a spill, but they do not know if and where this happens. The following query matches measurements passing a threshold and returning where and when this occurred.

```
MATCH (n:segment {SERIES :ec25:<a>}) WHERE a.value > 90000
RETURN n.vhas, a.timestamp, a.value;
```

The database returns one record: ("6033602", 2022-02-22T11:45Z, 99999.0). The *vhas* indicates on which segment in the river the spill would happen, the timestamp of the measurements indicates the time of the event, and we added the measured EC value. This type of query can also be used to set up data validation or find faulty measurements. Likely, this result could be a positive match for being a faulty measurement since it is exceptionally high.

**Query 6** The values of the measurements can be used to derive analytical results. In this case, the query retrieves a series
at a specific location and calculates a moving average window. Every possible pattern match is returned as a record, and each record selects 5 consecutive measurements.

```
MATCH (n:segment {SERIES IOW1:ec25:<a><b><c><d><e>})
WHERE a.timestamp < datetime("2022-01-01T04:00:00+0000")
RETURN a.timestamp, (a.value+b.value+c.value+d.value+e.value)/5;
```





**Table 4.** Records showing the average moving window returned by Query 6.

| a.timestamp | (a.value + b.value + c.value + d.value + e.value)/5 |
|---|---|
| 2022-01-01T00:00:00Z | 537.16 |
| 2022-01-01T00:15:00Z | 536.79 |
| 2022-01-01T00:30:00Z | 536.53 |
| 2022-01-01T00:45:00Z | 536.31 |
| 2022-01-01T01:00:00Z | 536.20 |
| 2022-01-01T01:15:00Z | 536.09 |
| 2022-01-01T01:30:00Z | 536.10 |
| 2022-01-01T01:45:00Z | 536.35 |
| 2022-01-01T02:00:00Z | 536.69 |
| 2022-01-01T02:15:00Z | 537.12 |
| 2022-01-01T02:30:00Z | 537.57 |
| 2022-01-01T02:45:00Z | 538.22 |
| 2022-01-01T03:00:00Z | 539.80 |
| 2022-01-01T03:15:00Z | 541.69 |
| 2022-01-01T03:30:00Z | 543.76 |
| 2022-01-01T03:45:00Z | 546.06 |

The results in Table 4 show the moving average for sensor IOW1 during the first 4 hours of the first of January.

**Query 7** Using more advanced graph patterns is useful to describe network structures where the location might be unknown. As this query shows, it looks for a path between two given locations in the network and retrieves the value measured on March 30th at midnight along possible paths. Specifically, the query looks for each EC25 labelled series on a possible path between segment 6031906 and segment 6033656.

```
MATCH (:segment {vhas: "6031906"})-[:flowsto*]->
            (n:segment {SERIES :ec25:<a>})-[:flowsto*]->
            (:segment{vhas: "6033656"})
       WHERE a.timestamp = datetime("2022-03-30T00:00:00+0000")
       RETURN n.vhas, a.value;
```

The database returns that at this point in time, there is a path with a series on segment 6042697, and the value there at this point in time is 7441.90.

**Query 8** In the IoW project, the water team of VITO established different use cases and models to analyse the Yser River and its water quality. One of the results is the IGOR tool, Desmet and Bauwel (2023), which is used to analyse electrical





conductivity on the Yser river. IGOR provides the ability to interpolate the sensor's measurements for any given location on
the river. This interpolation query is the final analysis step to demonstrate and test the database in this context. In the graph, it
should be possible to do the same interpolation for any location. Locations in the graphs are considered to be the nodes, and the
nodes correspond to segments. This would mean the interpolation can be done for each segment, with or without a sensor. This
matches the closest sensor upstream and downstream of the specified location. Subsequently, it interpolates the values for the
provided all points in time linearly to the distance between the sensors and the location. This means that, for a given location,
the entire time series is interpolated as long as both sensors have a value for the same moment in time.

```
MATCH up_path = (up:segment {SERIES IOW19:ec25:<x>})
            -[:flowsto*]->(i:segment {id:"14674"}),
       down_path = (i:segment {id:"14674"})-[:flowsto]->
            (down:segment {SERIES IMC_910040:ec25:<y>})
WHERE x.timestamp = y.timestamp
WITH *, REDUCE(s = 0, seg IN nodes(up_path) | s + seg.length) as up_distance,
     REDUCE(s = 0, seg IN nodes(down_path) | s + seg.length) as down_distance
RETURN  x.timestamp,
     (x.value*(1-(up_distance/(up_distance+down_distance))))
     + (y.value*(1-(down_distance/(up_distance+down_distance))))
```

The system's power lies in the fact that the sensors don't need to be known. With an additional subquery, the closest sensors can
be determined. The first `MATCH` part is the subquery that first explores the shortest path to a sensor and calculates the distance
to the sensors using two `WITH` clauses. This step must be separated from the rest because taking the minimum of a group of
matches is impossible if other fields are used. Using other fields with aggregation functions results in group-by clauses, similar
to SQL. To prevent this, the minimum is determined in the subquery. These minimum distances are subsequently used in the
second `MATCH` query to look up the series values and interpolate the EC values. The second part is the same query as before,
except that no sensor name is used, and instead, the distance of the matched sensor is compared to the minimum distance
obtained in the first part.

```
MATCH up_path = (up:segment {SERIES :ec25:<x>})-[:flowsto*]->
                (i:segment {id: "14674"}),
       down_path = (i)-[:flowsto*]->(down:segment {SERIES :ec25:<y>})
WITH REDUCE(s = 0, seg IN nodes(up_path) | s + seg.length) as up_distance,
     REDUCE(s = 0, seg IN nodes(down_path) | s + seg.length) as down_distance
WITH  min(up_distance) as up_min, min(down_distance) as down_min

MATCH up_path = (up:segment {SERIES :ec25:<x>})-[:flowsto*]->
                (i:segment {id:"14674"}),
```





```
            down_path = (i)-[:flowsto*]->(down:segment {SERIES :ec25:<y>}),
      WHERE x.timestamp = y.timestamp
WITH *,
          REDUCE(s = 0, seg IN nodes(up_path) | s + seg.length) as up_distance,
          REDUCE(s = 0, seg IN nodes(down_path) | s + seg.length) as down_distance
      WHERE up_distance = up_min AND down_distance = down_min
      RETURN i.id, x.timestamp, x.value, y.value,
(x.value*(1-(up_distance/(up_distance+down_distance))))
          + (y.value*(1-(down_distance/(up_distance+down_distance)))) as inter;
```

For the evaluation, the interpolation is executed for segment 14674 with all values available from January 1 until January 5.
The sensor IoW 18 is located on this segment, meaning all interpolated values can be compared to actual measured values.
The mean square root error can be determined from this, which is 44.61. With values in the range of 450 to 675, this error is

610 less than 10%. Considering that the sensors have an error margin of up to 10% for the measured EC values, this result can be
considered good. However, the interpolation performance is not the goal of this paper. The main takeaway of this use case is
that the proposed system can aid in solving relevant questions for transportation networks. That is what we considered proven
by being able to fulfil the same work as a dedicated programmed tool such as IGOR does.

## 6   Discussion

We acknowledge that it is possible to use existing temporal graphs to model and query transportation networks. There is work
from Kuijpers et al. Kuijpers et al. (2022) that models property graphs with categorical properties through time. Here, we
explicitly want to model time series, possibly with a high resolution, and we want to emphasise that the network is stable,
whereas the properties can take on many values through time. In addition, our approach exploits the fact that time series have
a natural order and uses this in graph pattern expressions. We discussed the advantages of our query language compared to the

620 procedures defined in addition to the existing query language. Nonetheless, procedures might need to be written because not
everything is expressible in the proposed query language. That is, expressing constraints on each pair of nodes in a path where
the path length is unbounded or expression constraints on each pair of measurements where the number of measurements is
unbounded is impossible.

The usage of labels in the experiment is a useful feature that was possible because the *Full Graph Model* is used in the

625 implementation. The implementation allows the labels of nodes to be used as labels for the time series. However, this use of
labels for time series properties is not fully explored and defined. This should be realised in future steps to standardise the
working, independent of the implementation method.

For each topology pattern, multiple measurement patterns can be valid in a time series. The number of matches in the end
result is the sum of all valid patterns in the time series for each valid spatial pattern. At most, each measurement is a valid match





for a time-series pattern, and then the number of topological matches needs to be multiplied by the number of measurements in the relevant time series. This can impact the efficiency. Based on the model and logic, here presented, this can now be studied more formally.

At the moment of writing, the system is demonstrated to and evaluated by expert researchers in hydrology and energy fields to evaluate the performance further and identify additional functionality. Therefore, the experiment shown should be seen as

a proof of concept. A performance analysis or benchmark is not conducted because the current implementation is not mature enough, and such a study would lead to false results. For example, the current translated queries rely on the transitive closure computation (the "*" operator) in Neo4j. However, earlier research has shown that this operator cannot handle path queries with unbound length as efficiently as other functions Bollen et al. (2021). However, we would like to indicate the usability based on this proof of concept. The database Neo4j database (version 4.2.3) was deployed on a server with Ubuntu 20.04, a two-core

system (Intel Xeon Gold 6136 CPU @ 3.00GHz) and 16Gb of memory. Queries 1 to 7 ran and retrieved results between 399 ms and 299 seconds (299358 ms), including transferring the records over the network to the local computer. Plus one-minute query time occurred for queries 3, 7, and 8. The translation of the queries from our language to full Cypher was performed between 5 ms and 382 ms. Again, we want to stress that all these results merely indicate the current state. The implementation is focused on functionality and not performance-oriented, nor are the timings conducted repeatedly. An objective study needs

to be conducted to validate the performance of an implementation that exhibits a higher technology readiness level.

## 7    Conclusions

Transportation networks are a recurrent research topic and can be modelled using property graphs. With the rise of the Internet of Things, measuring the status of the networks has become easier and the data volume has increased in terms of number of sensors and resolution of the measurements. These sensors produce time series that, ideally, should be included in the property

graphs. Temporal graphs focus on nodes and edges being valid at a certain time. In contrast, transportation networks rarely change their topology, but the measurements are properties of nodes or edges, where the values change through time. There also exist graph databases for storing property graphs, and time-series databases to store the measurements, but a combination, exploiting the characteristics, is missing. In this work, a property graph model with time series is proposed to provide a basic model. This model considers time series on nodes and edges together with the traditional properties as first class citizens. The

model is accompanied with a query language logic, exploiting the natural order on time of the measurements, to describe graph patterns that can be matched. In addition, the paper shows how this logic can be realised in graph query languages based on the Graph Pattern Matching Language (GPML). This all together, creates property graphs with time series, and leads to a tool set for querying transportation networks, or transforming them for more advanced processing. For the first time, it is possible to express patterns where there are constraints that take into account nodes, edges, properties, and measurements of time series,

that is timestamp and value pairs, at the same time. These can all depend on each other, providing the possibility to search and evaluate graphs on all these constraints at the same time. There is more work needed to formally study these graph with respect to complexity and evaluation, as the increased data size might impact the query evaluation. However, there are also possibilities



to exploit optimisations. For example, the new constraints can prune path searches earlier, or path searches can be limit the number of time series that need to be processed. The next step is, to realise this all in practice, by implementing the property graphs with time series in an actual database, and providing the proposed query language together with a query engine that process the queries. With this theory, a first step is set to store, process, and then also analyse transportation networks based on property graphs with time series. This in turn, can then support research concerned with these networks such as river networks, electricity networks and heat networks amongst other.

*Data availability.* The sensor data used in this paper is owned by Flemish Environmental Agency and made available on waterinfo.be[14]. The source network structure data is free and publicly accessible on the website of the Flemish Government download.vlaanderen.be[15]. A processed version of these two data sets, as it is used in the experiment section, can be found as a Neo4J database dump on OSF with DOI: 10.17605/OSF.IO/J9CN5, https://osf.io/j9cn5/.

*Competing interests.* There are no competing interests to declare.

*Acknowledgements.* The authors would like to thank Alejandro Vaisman and Valeria Soliani for fruitful discussions on property graphs with time-series data.

---

[14]https://www.waterinfo.be/Themas#item=water%20quality/physical%20parameters
[15]https://download.vlaanderen.be/catalogus?q=Vlaamse%20Hydrografische%20Atlas



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
