# Peer review of "Managing Data of Sensor-Equipped Transportation Networks using Graph Databases"

_Geoscientific Instrumentation, Methods and Data Systems, 2024_

## Author Response (AR1)

**W1.** Overall, at a lot of places the discussing the intuition behind the definitions could enhance the reading experience, making it easier for the reader to understand certain parts.

We agree that the theoretic section was dry and needed additional context. To prevent the definitions from being interleaved with explanations, we opted to add running examples that demonstrate and explain them. Two example queries are given and discussed at the end of the syntax section and semantic section. Both queries show how the definitions have to be applied and provide the reader with an additional explanation of how the different elements are used. Regarding W3, the usage of body, constraints and predicates is clarified this way too.

**W2.** The article can benefit from including existing definitions and notations that are extended for self-containment. For instance, the very basic GPML notation could be included in Sec 4.2 for a quick and smoother transition. Similarly, the article discusses extending Regular Property Graph Logic, but introduces its notations later in Sec 4.1.1 in GQL-TS, which may seem to the unassuming that it is a novelty/contribution of this article.

Thank you for this remark concerning the understandability and clarity of the contribution. We added a specific section introducing GPML to provide the smooth transition suggested. In addition, we tried to more clearly indicate for sections 4.1 and 4.2 which elements are existing work and which are novelties we added. Each time at the beginning of the section, we state how the reader can identify our contributions.

**W3.** Should include a discussion on the rules that are added to the grammar, the intuition and significance. May be a line around the predicate names, body and constraints in a similar manner would gain a better understanding of the usefulness of these early on in the article.

Thank you for this suggestion. We split the grammar rules into three logical groups, and for each, we explained the definition of grammar. The different elements of measurement patterns are now introduced, as well as how series patterns are written. The link with existing property descriptions is described separately afterwards. For the second part of this suggestion, we refer to our answer to W1, where we accommodate additional explanations for these parts.

**W4.** In Sec 5.1, authors proposed a variant of the discussed implementation which is a more consistent design. Including a comparison with it could result in a qualitative

analysis of the two data models, helping the end-user chooses the one better suited to their needs.

Thank you for pointing out that the difference between the model introduced in section four and the implementation model in 5 was not clearly explained. We adapted the introduction to section 5 to clarify the difference between the abstract graph model and the implementation model chosen to implement the abstract model.

**W5.** The experimental study is done on a real dataset from the Yser river basin, however, this data is very small and the scalability study is missing. Maybe a synthetic dataset could be created for this purpose? It would be good to have a comparison with the existing data models (e.g. Neo4j + Influx) to understand the significance of the proposed data model for such a small dataset.

Currently, we do not have such a data set ready to study the system's behaviour. We agree that this is additional work that has to be done. In addition, we think that it should include a survey of available database systems, database tuning options and a comparison of multiple implementation models. We do not have all these elements ready at the moment, and a less elaborate comparison would fall short of delivering a valid comparison leading to a conclusion that can be easily disproved by a new experiment. At this moment in time, the contribution of the design of the model and the additional research mentioned above is a part of what we want to do in the future.

**W6.** Avoid using similar symbols for labels and predicates, it is confusing for the readers. Similarly, it is not a good idea to abuse notations, especially since there are many definitions involved and it is not always clear by context if the notation is abused.

Thank you for mentioning this observation. We use the term predicates in the theory to identify the logical predicate object of the rules, which has additional meaning to the labels in the graph itself. Predicate names align with the label names in practice, but to differentiate between the two and to keep naming consistent with other theories, we like to keep this naming convention. Using the label as naming for a predicate in the theory would lead to confusion later on where for example, the sentence would become: 'Then, we define the semantics of this label ell to be …' instead of 'Then, we define the semantics of this predicate ell to be' where we asses that the semantics of a label is more confusing.

**W7.** In Sec 4.1.2, use numbered bullets for discussing semantics of predicates.

Thank you for this suggestion. We understand that the deeply nested bullet lists are not easy to read. The third level of nesting was removed to make it easier to follow. In addition, we updated specific parts to an enumeration instead of an itemisation for parts where the items form a conjunction. This should give the reader a number of conditions which need to be fulfilled. The itemisation is kept for other parts to indicate that we dealing with a disjunction.

**W8.** The query for example 4.4 follows the grammar format before it is introduced, probably an unintentional mistake.

Thank you for pointing this out. This is indeed a mistake where we forgot to alter the notation of the query. We updated this query with the correct notation.

**W9.** Typographical errors:
- – Irregular citation format - e.g. Bollen Bollen (2022) instead of Bollen (2022). Many such irregularities.
- – Line 274: exits → exists.
- – Line 284: Instead of . . .mk, ∗,mk+1 . . . using k' instead of k + 1 would clarify the Kleene star operation better in the constraint.
- – Line 499: "N additional constraints on . . .", if not a typo please write more specifically.

Thank you for these corrections which we did not notice before sending in the draft. All of them are addressed. Except for suggestion 3, we still prefer the notation with k+1 because using k' entails additional requirements that need to be written out later, which would make the definition longer.

**C1. Line 30: Could you find more links for this problem?**

We added additional resources that support the claims on line 30.
Gamper and Dignös provide a paper titled "Processing Temporal and Time Series Data: Present State and Future Challenges", which support the statements in the following sentences:

"However, managing time series in graph databases is not trivial. Additional data structures are required to store the time series, and the existing query languages do not support time series to express constraints."

**C2. Do you have a stability statistics? How much data was lost? What are the statistics on system hangs? How accurate is the binding of sensor data to system time?**

This is an interesting comment, thank you.

The data gathering itself is not considered a part of this work. Therefore, we did not include the information on the amount of data lost or related statistics.
However, we can report that 22% of data is not present in the current data set compared to the ideal data set.
Most of this missing data are values removed because they are registered during sensor cleaning and recalibration moments. Some other points are missing due to some technical issues.
The time of the measurements is registered at the sensor and then sent to the data storage. The measured values and timestamps are verified and cleaned to obtain time series with the same timestamp at a 15-minute resolution in the storage system. We use this cleaned data in our experiments and consider those as facts that are assumed true at that time. Therefore, we do not report on the relation between valid and system time.
We clarified both points additionally at the beginning of section "5.3 Experiment".

The systems performances we could test are reported in the discussion section. We understand that the data is limited and incomparable to other benchmark studies. Additional development and extensive tests need to be conducted to have a qualitative performance result. We do not have these elements ready to study the system's behaviour at this point. We agree that this is additional work that has to be done in the future.